# Weight Development and Growth Curves of Grazing Santa Inês Sheep Supplemented with Concentrate in the Pre-Weaning Phase

**DOI:** 10.3390/ani14121766

**Published:** 2024-06-12

**Authors:** Rodrigo Ferreira da Silva, Pedro Henrique Cavalcante Ribeiro, Yasmin dos Santos Silva, Maria Alice de Lima Soares, Cláudio Vaz De Mambro Ribeiro, Adriano Henrique do Nascimento Rangel, Marcelo de Andrade Ferreira, João Virgínio Emerenciano Neto, Stela Antas Urbano

**Affiliations:** 1Academic Unit Specialized in Agrarian Sciences, Federal University of Rio Grande do Norte, Macaíba 59280-000, RN, Brazil; rodrigozootecnia22@gmail.com (R.F.d.S.); pedrohcrib@gmail.com (P.H.C.R.); yasminsszoo@gmail.com (Y.d.S.S.); mariaalicesoares07@gmail.com (M.A.d.L.S.); adrianohrangel@yahoo.com.br (A.H.d.N.R.); joao.emerenciano@ufrn.br (J.V.E.N.); 2Department of Animal Science, Federal University of Bahia, Salvador 40170-110, BA, Brazil; cvdmribeiro@gmail.com; 3Department of Animal Science, Federal Rural University of Pernambuco, Recife 52171-900, PE, Brazil; marcelo.aferreira@ufrpe.br

**Keywords:** modeling, performance, regression equation, sheep meat

## Abstract

**Simple Summary:**

Weaknesses in the sheep meat production chain are largely related to non-specialized systems in the production of this product, which results in the supply of low-quality meat to the consumer market. In this regard, the intensification of animal management is becoming increasingly important. In this study, we evaluated the growth curves of Santa Inês lambs from the suckling phase to weaning to monitor their weight development according to sex and type of birth. Simple and multiple regression equations were evaluated to estimate live weight at weaning. We also found that the best equations for estimating the weight of lambs, regardless of sex and type of birth, were models with more than one biometric measure. However, from a practical point of view, body length and barrel girth can be efficiently used to estimate the weight of Santa Inês sheep at 84 days of age.

**Abstract:**

Monitoring weight development is essential for decision-making and assessing the effectiveness of management strategies. However, this practice is often hindered by the lack of scales on farms. This study aimed to characterize the weight development and growth curves of male and female Santa Inês lambs from birth to weaning, managed on pasture with creep-fed concentrate supplementation. Data from 212 lambs during the pre-weaning phase were analyzed. The animals were weighed every seven days to evaluate total weight gain and average daily gain. Biometric measurements were taken every 28 days. Mixed models were used to assess the effects of sex and birth type on birth and weaning weights. Simple and multiple linear regression models were employed to estimate live weight using biometric measurements. The non-linear Gompertz model was utilized to describe weight development and formulate growth curves. Results were considered significant at *p* < 0.05. An interaction effect between birth type and sex (*p* < 0.05) was noted for birth weight, with the lowest weight observed in twin-birth females (2.96 kg) and the highest in single-birth males (3.73 kg) and females (3.65 kg) (*p* > 0.05). Birth type significantly influenced average daily gain, total weight gain, and weaning weight (*p* < 0.05). The Gompertz model accurately depicted the growth curves, effectively describing the weight development. Pearson’s correlation coefficients between biometric measurements and weight were positive and significant (*p* < 0.05), ranging from 0.599 for hip height to 0.847 for heart girth. Consequently, the simple and multiple regression equations demonstrated high precision in predicting weaning weight. In conclusion, twin-birth lambs receiving concentrate supplementation via creep-feeding and managed on pasture showed different developmental patterns compared to single-birth lambs under the same conditions. The Gompertz model proved effective for monitoring development during the pre-weaning phase. All simple and multiple linear regression models were effective in predicting weaning weight through biometric measurements. However, for practical application, the model incorporating two measurements—body length and abdominal circumference—is recommended.

## 1. Introduction

The frequent sale of carcasses not derived from specialized meat production systems often leads to the market being supplied with products that fail to meet the quality standards demanded by consumers, such as tenderness, color, and fat percentage. These qualities are directly linked to the age at which the animal is slaughtered, necessitating accelerated production cycles to enhance the quality of sheep meat. It is hypothesized that enhancing the quality of meat reaching consumers could increase per capita consumption of this product [1]. Therefore, feeding strategies aimed at shortening the production cycle can contribute to and improve these quality parameters.

Monitoring lamb weight throughout its development is crucial for controlling animal health and ensuring the efficiency of the production system. However, Santos et al. [2], noted that such monitoring is infrequent, as many producers lack the necessary technologies or infrastructure, leading to significant challenges. Thus, simple methods like biometric measurements can be adopted to facilitate weight monitoring and address performance issues, proving to be a valuable tool in predicting animal weight [3,4].

Tracking animal growth and development via growth curves also aids in precise decision-making, particularly in selecting animals and planning early feeding programs [5]. These curves are constructed using non-linear functions that utilize extensive datasets and easy-to-understand biological parameters, delineating the relationship between animal age and weight [6]. The Gompertz and Von Bertalanffy models are frequently used to describe the development of meat sheep, offering the best fit to biological evaluation parameters [7]. Among these, the Gompertz model shows particularly high goodness of fit for Santa Inês sheep in the pre-weaning phase [8].

In pasture-based production systems, intensifying management techniques to reduce slaughter age is partially constrained because weight gain per animal and per area is heavily influenced by pasture availability, stocking capacity, and forage quality [9]. This limitation has spurred research into nutritional strategies that can be integrated with pasture use to boost productivity. This includes the strategy of providing concentrate supplements, previously emphasized by Miorin et al. [10] as essential for the sustainability of pasture-based systems and by Gurgel et al. [11] as a solution to performance issues in animals managed on cultivated tropical pastures.

In view of the above, it is inferred that there is a need to validate the technologies generated with the purpose of increasing the efficiency of pasture-based production systems, without compromising sustainability and economic viability. Therefore, it is considered possible that offering a concentrate supplement to lambs managed on pasture can enhance weight development during the pre-weaning phase and contribute to reducing the age at slaughter, providing a valid and effective nutritional management strategy, in addition to an alternative to the use of scales capable of contributing to the consolidation of the sheep meat production chain. With this, the objective was to characterize the weight development of lambs and ewe lambs of the Santa Inês breed, from birth to weaning, managed on pasture and receiving concentrated supplementation via creep-feeding.

## 2. Material and Methods

### 2.1. Experimental Locations and Animals

Data from 212 sheep, aged 1 to 84 days, were used. They comprised all 108 males and 104 females from four birth seasons: 2018, 2019, 2020, and 2021. These sheep were part of the experimental herd at the Sheep Teaching and Research Laboratory, located in the Specialized Academic Unit in Agricultural Sciences (UAECA) at the Federal University of Rio Grande do Norte (UFRN). Throughout the data collection period, the animals were managed under the same conditions. They were kept on pasture planted with Megathyrsus maximum cv. Massai, covering a total area of 0.96 ha (9600 m²), which was equipped with drinkers and a trough for mineral supplementation. Supplementary concentrate was provided in a management shed, which also had drinkers and feeders.

### 2.2. Animal Management and Data Collection

The experimental animals received immediate postnatal care. After colostrum ingestion and maternal acceptance, they were weighed and identified, and their navels were treated. The sex of each lamb and the type of birth (single or twin) were recorded. The lambs remained with their mothers until 14 days old, when controlled feeding started, leading up to weaning at 84 days. During the day, while the mothers grazed, the lambs stayed in a Massai paddock from 07:00 to 15:00. After this time, they rejoined their mothers in collective pens, where they spent the night. The pens had feeders for the dams and drinkers with water available ad libitum. Additionally, a separate feeding space (creep-feeding) was created, allowing lambs access to specific concentrate.

During the pre-weaning phase, the lambs were fed ad libitum with a concentrate that contained 23.6% crude protein and 3340 Mcal DE/kg. This concentrate was composed (dry-matter basis) of 60.1% ground corn grain, 36.9% soybean meal, 2.5% mineral mix, and 0.5% common salt. Solid feed was provided in private troughs from the first hours of the lambs’ lives to encourage solid feed intake.

Weight development was evaluated by weighing the lambs every seven days, from birth to 84 days old, always before they were sent to pasture. These data were used to calculate total weight gain and average daily weight gain. Total weight gain was calculated by subtracting the birth weight from the weight at 84 days. Average daily gain was derived by dividing the total weight gain by the number of days (84).

Biometric measurements were taken at 84 days using a measuring tape graduated in centimeters and following the methodology described by Cezar and Sousa [12]. The measurements included hip height (HH), withers height (WH), abdominal circumference (AC), heart girth (HG), chest width (CW), rump width (RW), and body length (BL), measures that can be easily taken in the field by any breeder.

### 2.3. Data Processing and Statistical Analysis

The collected data were organized into electronic spreadsheets and adjusted for weights at birth and at weaning, which occurred at 84 days. A mixed model in a randomized block design was used to examine the effects of sex, birth type, and their interaction on weight gain variables, with the year serving as the blocking criterion. The model used is as follows:Yijk = µ + Si + Bj + SBij + yk + eijk,
where Yijk = dependent variable, µ = overall mean, Si = fixed effect of sex (i = 1.2), Bj = fixed effect of birth type (j = 1, 2), SBij = interaction effect between sex and birth type, yk = random effect of year (k = 1, 2, 3), and eijk = residual error.

Pearson correlation analysis was used to assess the relationship among biometric measurements. Simple and multiple linear regression models were tested to estimate live weight based on biometric measurements as independent variables.
Simple linear regression: Y= b0 + b1X1
Multiple linear regression: Y = b0 + b1X1 + … + bnXn,
where Y = dependent variable, b0 = intercept, b1X1 = estimated regression coefficient (b1) for the first independent variable (X1), … = the same for all independent variables tested, and bnXn = estimated regression coefficient (bn) for the last independent variable (Xn).

Von Bertalanffy, another non-linear model was also tested to derive growth curves for each category. But the model that best fit the data was the Gompertz model, with three parameters:W = ae−be−cI + ε,
where W = animal weight (kg), a = asymptote, b = growth rate, c = inflection point, I = animal age (days), and ε = residual error.

## 3. Results and Discussion

A significant interaction effect (*p* < 0.05) was observed between sex and birth type on birth weight. The lowest average birth weight was found in females from twin births, at 2.96 kg. The highest birth weights were seen in males (3.73 kg) and females (3.65 kg) from single births (*p* > 0.05) (Table 1).

There was a significant effect (*p* < 0.05) of birth type on total average daily gain (ADG), total weight gain (TWG), and weaning weight. The difference in total ADG between the groups was 38 g/day, which translated into a 3.2 kg difference in TWG and a 3.7 kg difference in weaning weight (Table 2). However, when birth type is not considered, males and females had similar total ADG, at 161 and 168 g/day, respectively. No statistical differences were observed in TWG or weaning weight (Table 2).

The growth curves derived from the Gompertz model, categorized by sex and birth type, demonstrate a strong fit and align with observations in the pre-weaning development of Santa Inês breed sheep. As evidenced in the performance results (Table 1), the growth pattern appears to be mainly influenced by the type of birth. Notably, males and females from single births had similar growth patterns, while males from twin births showed superior development compared to females from twin births (Figure 1).

The Gompertz model provided a high degree of accuracy and potential for use in monitoring weight development and aiding in nutritional management during the pre-weaning phase (Figure 2).

According to the model evaluation criteria—weight at maturity (a), growth rate (b), and inflection point (c) (Table 3)—males and females from single births are more precocious and are preferable in meat production systems for both reproduction and finishing. Although females from twin births have higher growth and maturity rates than males from twin births, these rates do not necessarily suggest better performance, but rather earlier sexual maturity.

The model parameters support the earlier observation that single-birth males and females outperform twin-birth animals. It is worth noting that the weight at maturity for females from single births was lower compared to other categories, and the inflection point for males from twin births was the highest.

Weights had positive correlations with all biometric measurements, ranging from 0.599 with HH to 0.847 with HG, indicating that these measurements are effective for predicting body weight at weaning (Table 4).

Based on biometric traits, it was possible to create simple and multiple linear equations to estimate the weaning weight at 84 days for Santa Inês sheep (Table 5). All equations had high coefficients of determination. However, equations with two or more measurements had coefficients of determination greater than 80%, suggesting that body length and abdominal circumference can be used to propose an equation for weight estimation (adjusted R² = 0.824).

## 4. Discussion

The average birth weights of the animals evaluated in this study ranged from 3 to 4 kg, consistent with the typical range for Santa Inês sheep reported in the literature [3,13,14]. However, an exception is noted for females from twin births, who had lower birth weights. These average weights are generally considered satisfactory, as birth weight can impact lamb development. As Peruzzi et al. [14], highlighted, lower birth weights are associated with decreased survival rates, which can affect the number of kilograms of lambs weaned per year, ultimately impacting meat production and the revenue for the production system.

Several factors can significantly influence birth weight, including the type of birth, ewe nutrition, breed, and sex. Although the nutritional status of the dams was not evaluated in this study, the observed birth weights within the typical range suggest that the dams were likely well-nourished during gestation. Kamjoo et al. [15] noted that environmental factors could interfere with birth weight, particularly those related to the ewes, such as nutritional management, which Torres et al. [16] highlighted as an important aspect of production systems that could compromise results.

The birth weight of animals from twin births and their subsequent performance are entirely dependent on the competition that takes place during the intrauterine phase, with pre-birth performance tied to milk production [17]. Even in the intrauterine phase, twin-birth males may receive more nutrients than females, possibly explaining their higher birth weights. Additionally, testosterone acts on satellite cells of the myogenic lineage concurrently with muscle hypertrophy, modifying the number of myonuclei, somatic cells, and adipose mass [18]. From this perspective, it is possible that males exhibit greater growth speed, better feed efficiency, and higher muscle tissue deposition [19]. This process is key for the development of twin-birth animals but does not influence those from single births, where nutrient transfer via the placenta is exclusive and without competition.

Despite differences in birth weight, animals from single and twin births exhibited similar performance during the pre-weaning phase, as evidenced by performance data (Table 2). This observation can be explained by the limited impact of sexual hormones at this age, which do not significantly influence the development of single-birth animals [20]. Torres et al. [16] indicated that average daily gain is a key factor for achieving higher weaning weights. Concentrate supplementation can be an effective strategy to attain these objectives. Soares et al. [21] reported that Santa Inês breed animals from single births, weaned at 90 days and given concentrate supplementation, can achieve gains of 157 g/day. Catto et al. [22] found similar gains of 160 g/day in the same category. In our studies, the average daily gain was 26 g higher than those reported for the same category, resulting in an average weaning weight of 19.1 kg.

Weaning weight is an important indicator of the success of an animal in the finishing phase. In this study, animals born from single births had higher average weaning weights compared to those from twin births (Table 2). No significant differences were observed in weaning weights between male and female animals (*p* > 0.05). These results were higher than the 14.02 kg weaning weight reported by Soares et al. [21], for Santa Inês breed animals from single births, weaned at 90 days of age. However, they were similar to the 17.7 kg described by Catto et al. [22] in animals also weaned at 90 days with concentrate supplementation on Brachiaria grass pastures. In contrast to these findings, Farias Jucá et al. [23] reported no significant differences in weaning weight based on birth type, but their study involved weaning at 112 days. This suggests that animals from twin births may exhibit compensatory gains when weaned at a later age.

The choice of a model for monitoring animal development is subjective and varies depending on existing models. Teixeira Neto et al. [7] emphasize that the chosen model should represent the development of the largest number of animals. In practice, a single model might overestimate some variables, such as birth and weaning weight [24]. However, in this study, the Gompertz curve closely matched the observed average weight from birth to weaning, suggesting that weight estimates at maturity are adequate if nutritional and health requirements are perfectly met.

Weight at maturity is linked to growth rate, with higher growth rates generally translating into lower weights at maturity [5]. Categories with lower growth rates take longer to reach maturity, which is undesirable in meat production. At the same time, a high growth rate indicates earlier sexual development, which can increase the production cycle turnover [25,26]. The growth rate was higher for males and females from single births (Table 3), as well as females from twin births (Table 3). Even though males from twin births showed better performance than females, it is inferred that they might require more time to reach an ideal slaughter weight. Santos et al. [25] reported a similar growth rate (0.02) for the Gompertz model applied to Santa Inês and Ile de France breed animals. The growth rate seems more influenced by birth type, with a greater effect on males from twin births, whose development may be delayed.

At 84 days of life, the growth rate accelerates due to increased muscle deposition compared to other tissues contributing to empty weight [25,27]. This changes as animals approach maturity, when adipose tissue deposition increases [28]. Females from twin births were estimated to have lower weight at maturity than other categories, likely due to their lower weaning weight and earlier sexual maturity, reaching puberty between 150 and 300 days [29].

Biometric measurements are highly correlated with live weight and can be useful for predicting animal weight [30]. The proposed equations support this information, but they indicate that the highest correlations between the body weight of a lamb and biometric measurements are not always the most reliable for predicting weight at weaning. Koritiaki et al. [31] suggest that multiple linear regressions using more than one trait are better for predicting animal weights than simple equations. Generally, the more measurements included, the more accurate the weight estimation. Our results showed that using heart girth or abdominal circumference in simple equations accounts for 70% of the power to predict weight at 84 days of age. Nonetheless, when body length is combined with abdominal circumference, the reliability of the prediction increases to over 82%.

## 5. Conclusions

Twin-birth lambs receiving concentrate supplementation through creep-feeding and managed on pasture have lower growth rates than single-birth lambs. However, when weaned at 84 days, their weight is satisfactory for intensifying meat production systems and shortening the finishing period. The Gompertz model can be employed to predict animal development and support decision-making within the production cycle. Linear equation models, both simple and multiple, are effective for predicting weaning weight using biometric measurements. From a practical perspective, equations with two measurements (body length and abdominal circumference) proved to be more efficient.

## Figures and Tables

**Figure 1 animals-14-01766-f001:**
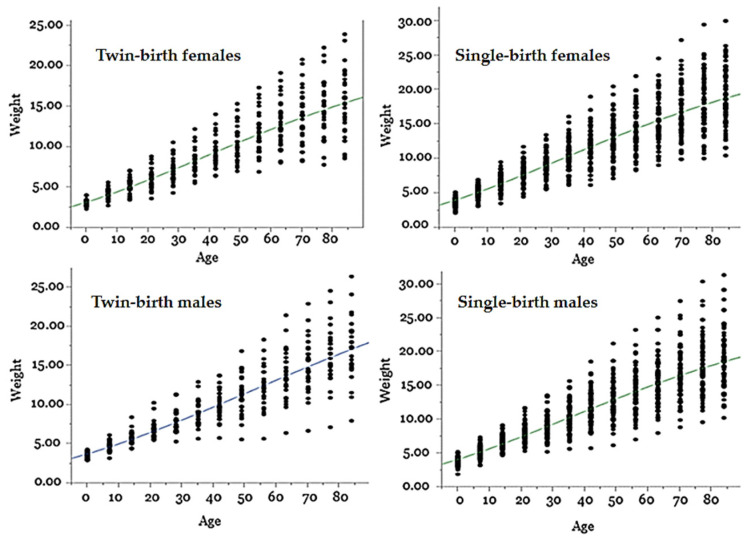
Growth curves of male and female Santa Inês sheep from single or twin births, adjusted according to the Gompertz model, with three parameters.

**Figure 2 animals-14-01766-f002:**
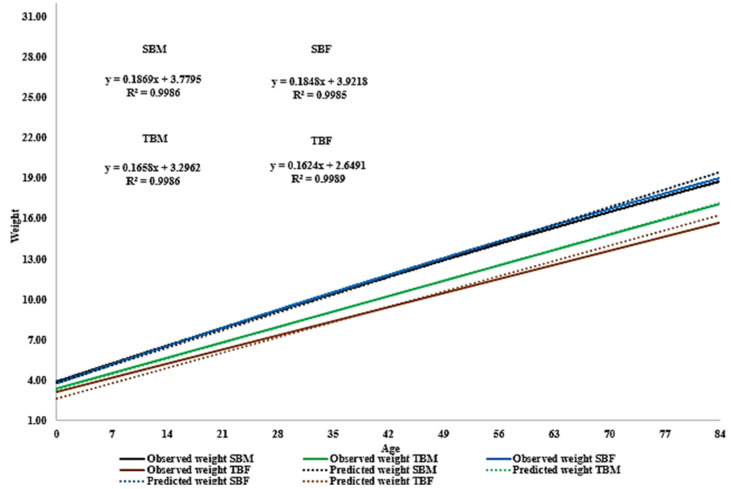
Growth curve adjusted according to the Gompertz model, superimposed on the observed average weight of male and female Santa Inês sheep from single or twin births. SBM = single-birth males; TBM = twin-birth males; SBF = single-birth females; TBF = twin-birth females.

**Table 1 animals-14-01766-t001:** Interaction effect between sex and birth type on the birth weight of Santa Inês lambs.

Variable	Male	Female	SEM	*p*-Value
Single	3.73 ^Aa^	3.64 ^Aa^	0.154	1.000
Twin	3.38 ^Ab^	2.95 ^Bb^	0.171	0.007
SEM	0.164	0.162	-	-
*p*-value	0.018	0.001	-	-

Different uppercase letters in the same row indicate significant differences (*p* ≤ 0.05) according to the *t* test. Lowercase letters in the same column indicate significant differences (*p* ≤ 0.05) according to the F test. SEM = standard error of the mean.

**Table 2 animals-14-01766-t002:** Effect of birth type on weight, total average daily gain, total weight gain, and weaning weight of Santa Inês lambs.

	Birth Type		SEM	*p*-Value
	Single Birth	Twin Birth		
Total average daily gain (g/day)	183	145	0.015	<0.001
Total weight gain (kg)	15.4	12.2	1.285	<0.001
Weaning weight (kg)	19.1	15.4	1.365	<0.001
	Sex			
	Male	Female		
Total average daily gain (g/day)	161	168	0.015	0.212
Total weight gain (kg)	14.1	13.5	1.280	0.212
Weaning weight (kg)	17.7	16.8	1.365	0.089

SEM = standard error of the mean; *p* ≤ 0.05 is described as significant according to the *t* test.

**Table 3 animals-14-01766-t003:** Estimates of weight at maturity (a) and growth rate (b) and inflection point (c) of the growth curve of single- and twin-birth males and females according to the Gompertz model.

Variable	a	b	c
Twin-birth males	31.9264	0.0147	51.2897
Single-birth males	27.5335	0.0187	33.8271
Twin-birth females	23.8523	0.0183	37.7444
Single-birth females	27.2649	0.0195	32.8854

**Table 4 animals-14-01766-t004:** Pearson correlation coefficients between weight and biometric measurements in Santa Inês lambs.

	Weight	BL	WH	HH	HW	RW	HG
BL	0.788 ***	-	-	-	-	-	-
WH	0.744 ***	0.601 ***	-	-	-	-	-
HH	0.599 ***	0.391 ***	0.759 ***	-	-	-	-
HW	0.643 ***	0.422 ***	0.415 ***	0.521 ***	-	-	-
RW	0.713 ***	0.616 ***	0.435 ***	0.314 *	0.587 ***	-	-
HG	0.879 ***	0.712 ***	0.718 ***	0.414 ***	0.504 ***	0.646 ***	
AC	0.843 ***	0.597 ***	0.626 ***	0.481 ***	0.556 ***	0.602 ***	0.820 ***

BL = body length; WH = withers height; HH = hip height; HW = hip width; RW = rump width; HG = heart girth; AC = abdominal circumference; * ≤0.05; *** ≤0.001.

**Table 5 animals-14-01766-t005:** Simple and multiple regression equations for predicting the weight of Santa Inês lambs and ewe lambs weaned at 84 days based on biometric measurements.

Number of Measurements	Equation	Adjusted R²
7	WW = −33.089 + (0.234 × BL) + (0.103 × WH) + (0.113 × HH) + (0.269 × HW) + (0.284 × RW) + (0.116 × HG) + (0.182 × AC)	0.885
5	WW = −33.038 + (0.235 × BL) + (0.365 × RW) + (0.148 × HG) + (0.189 × AC) + (0.226 × HH)	0.881
4	WW = −32.524 + (0.247 × BL) + (0.447 × RW) + (0.276 × WH) + (0.253 × AC)	0.869
4	WW = −27.656 + (0.269 × BL) + (0.475 × HW) + (0.157 × HG) + (0.211 × AC)	0.863
3	WW = −32.084 + (0.280 × BL) + (0.288 × HH) + (0.347 × HG)	0.833
3	WW = −33.624 + (0.368 × BL) + (0.216 × HH) + (0.309 × AC)	0.852
2	WW = −27 + (0.398 × BL) + (0.355 × AC)	0.824
2	WW = −22.031 + (0.318 × HG) + (0.271 × AC)	0.787
1	WW = −17.303 + (0.560 × HG)	0.717
1	WW = −19.842 + (0.509 × AC)	0.707

WW = predicted weaning weight; BL = body length; WH = withers height; HH = hip height; HW = hip width; RW = rump width; HG = heart girth; AC = abdominal circumference.

## Data Availability

Data are contained within the article.

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
