# Peer review of "Weight Development and Growth Curves of Grazing Santa Inês Sheep Supplemented with Concentrate in the Pre-Weaning Phase"

_animals, 2024, doi:10.3390/ani14121766_

Round 1

Reviewer 1 Report

Comments and Suggestions for Authors

General comments

This is a solid study, with an adequate sample size; I am not qualified to comment on the statistics, but the analysis seems good.

Simple summary

Line 19 « The Gompertz model showed the highest accuracy in tracking the weight development of sheep …” you say later in …2.3. Data processing and statistical analysis

Line 145  “Non-linear models were also tested to derive growth curves for each category. The model that best fitted the data was the Gompertz model,” I do not see which other models were tested; I suggest you say here which you tested. 

Results

Table 2 I expect « 168 » should read « 16.8 »  

Fig. 1     I suggest for the legend you get rid of “A = twin-birth females. B = single-birth females; C = twin-birth males; D = single-birth males.” by  putting the info directly on the Fig., e.g.  “Twin-birth females” or “TBF” to replace “A”

Table 4 Bottom line unreadable in this file.

You say “males and females from single births are more precocious and are preferable in production systems for both reproduction and finishing.” I would rephrase because the preferable types will depend on the objective of a production system, so this phrase is not true for all such systems..

Discussion

In the Introduction you say in Line 81…“validate the technologies  generated with the purpose of increasing the efficiency of pasture-based production systems, without compromising sustainability and economic…”

In the Discussion all I can see about validation is …”These average weights are generally considered satisfactory,…”; I suggest you say how your paper helps the validation process. 

Editorial

Line 177 “developm ent compared” the gap needs correcting.

Line 239               “breast milk production” I would drop “breast” since milk can come only from the udder.

Author Response

Dear reviewer, we appreciate your care with our material and inform you that 100% of your requests were met and are marked in the text of the manuscript.

Line 19 « The Gompertz model showed the highest accuracy in tracking the weight development of sheep …” you say later in …2.3. Data processing and statistical analysis. (WE CORRECTED BY REMOVING THE TEXT)

Line 145 “Non-linear models were also tested to derive growth curves for each category. The model that best fitted the data was the Gompertz model,” I do not see which other models were tested; I suggest you say here which you tested. (WE CORRECTED BY INCLUDING THE MODEL NAME)

Table 2 I expect « 168 » should read « 16.8 » (WE CORRECTED)

Fig. 1 I suggest for the legend you get rid of “A = twin-birth females. B = single-birth females; C = twin-birth male; D = single-birth evils.” by putting the info directly on the Fig., e.g. “Twin-birth females” or “TBF” to replace “A” (WE MADE THE ADJUSTMENT EXACTLY AS SUGGESTED)

Table 4 Bottom line unreadable in this file. (WE CANNOT MOVE THE FORMATTING AND WE WILL NEED HELP FROM THE JOURNAL TEAM TO RESOLVE THE OVERLAPPING TEXTS)

You say “males and females from single births are more precocious and are preferable in production systems for both reproduction and finishing.” I would rephrase because the preferable types will depend on the objective of a production system, so this phrase is not true for all such systems.. (CORRECT)

Line 177 “developm ent compared” the gap needs correcting. (OK, IT HAS BEEN CORRECTED. THANK YOU!))

Line 239 “breast milk production” I would drop “breast” since milk can eat only from the udder. (WE CORRECTED BY DELETING THE WORD "BREAST")

Reviewer 2 Report

Comments and Suggestions for Authors

This study evaluated the impact of concentrate supplementation on the weight development of grazing Santa Inês lambs from birth to weaning. Using the Gompertz model, it found that twin-born lambs had lower growth rates but suitable weaning weights for meat production. Regression equations using body length and abdominal circumference effectively predicted weaning weight.

1.The study includes only 212 lambs, so increasing the sample size would enhance the representativeness and statistical reliability of the results.

2.The study does not evaluate the nutritional status of the ewes during gestation, which significantly affects lamb birth weight and growth. Including this variable would provide more comprehensive results.

3.The study uses only seven biometric parameters (hip height, withers height, abdominal circumference, heart girth, chest width, rump width, and body length). Consider adding more parameters, such as leg length or back width, to improve weight prediction accuracy.

4. The article mentions that environmental factors may affect the results but does not specifically control or record these factors. Detailed recording and control of variables like temperature, humidity, and pasture quality are recommended.

5.The resolution of Figure 1 is too low, and the image is blurry. It needs to be replaced with a higher-resolution version.

Comments on the Quality of English Language

The language still needs further revision and improvement.

Author Response

Dear Reviewer,

We appreciate your care with our manuscript and we understand all your questions. We've worked hard to improve the quality of the figures and hope they look more satisfactory.

Regarding points 1 and 3 of your comments, we regret the lack of ability to adjust. We work with a database formed during 4 birth seasons of our herd. As the data was already collected, unfortunately we do not have additional animals or measurements.

Regarding point 4, we have the historical rainfall series. Would it be in good taste for us to add it?

Reviewer 3 Report

Comments and Suggestions for Authors

The manuscript titled "Weight development and growth curves of grazing Santa Inês sheep supplemented with concentrate in the pre-weaning phase" presents an insightful study on the growth patterns of Santa Inês lambs. The research is well-structured, and the data collected is extensive, providing valuable insights into the weight development and the effectiveness of different biometric measurements in predicting weaning weight. The use of the Gompertz model is appropriate and the statistical analysis is robust. However, there are several areas where the manuscript could be improved to enhance clarity, depth, and overall quality.

1. Introduction Section: Expand on the significance of the study by detailing the implications of improved weight monitoring for sheep farming. Clearly state the hypotheses being tested.

2. Materials and Methods: Include more details about the environmental conditions of the experimental site. Specify the criteria used for selecting the 212 sheep.

3. Clarify the method used for taking biometric measurements, especially the tools and techniques. Explain the rationale for choosing specific biometric measurements.

4. Provide a more in-depth discussion on how the findings compare with previous studies. Discuss the practical implications of the findings for sheep farmers in more detail.

5. Conclusion: Summarize the key findings more succinctly. Highlight the potential for future research based on the study's findings.

Author Response

Dear Reviewer,

We appreciate the care taken with our manuscript and, above all, the positive comments made on our work.

We made all the adjustments requested in your review and they are highlighted in the text using the track and change tool. It is important to clarify that we work with a database built over 4 birth seasons of our experimental herd. 212 animals is the total number of animals born in these seasons, which explains the number. We changed the methodology and conclusion as suggested.

We hope to have met your expectations and remain at your disposal for further clarification.